# Allied Health Professions Accreditation Standards for Work Integrated Learning: A Document Analysis

**DOI:** 10.3390/ijerph20156478

**Published:** 2023-07-31

**Authors:** Merrolee Penman, Jacqueline Raymond, Annora Kumar, Renae Y. R. Liang, Karen Sundar, Yvonne Thomas

**Affiliations:** 1Curtin School of Allied Health, Faculty of Health Sciences, Curtin University, Bentley, WA 6102, Australia; 2School of Health Sciences, Faculty of Medicine and Health, The University of Sydney, Sydney, NSW 2006, Australia; jacqueline.raymond@sydney.edu.au; 3Medical School, Health and Biomedical Sciences, The University of Western Australia, Crawley, WA 6009, Australia; 22711021@student.uwa.edu.au; 4School of Medicine, The University of Notre Dame, Fremantle, WA 6160, Australia; renae.liang@my.nd.edu.au; 5Curtin Medical School, Faculty of Health Sciences, Curtin University, Bentley, WA 6102, Australia; karen.sundar@student.curtin.edu.au; 6School of Occupational Therapy, Otago Polytechnic, Te Pūkenga, Dunedin 9054, New Zealand; yvonne.thomas@op.ac.nz

**Keywords:** allied health, work integrated learning, health professions education, clinical placements, accreditation standards

## Abstract

A key role of allied health (AH) professional regulatory and professional bodies is to ensure that AH education programs provide work-integrated learning (WIL) opportunities for students. The requirements are outlined via the respective profession’s educational accreditation standards. Although a significant component of the AH professional degrees, researchers have not explored how standards specific to WIL are developed, nor how WIL might be conceptualised through the standards. This study explored how WIL is conceptualised through comparing the WIL education standards across Australian AH professions. Using a non-experimental explanatory mixed-methods research design, a document analysis of Australian education program accreditation standards (and associated documents) for 15 AH professions was undertaken. Data analysis included inductive textual and thematic analyses to compare AH professionals’ conceptualisation of WIL. This study found a high degree of variation in how AH professions describe WIL. While there was a common requirement for students to demonstrate competency in WIL, requirements for WIL quantity, assessment and supervision varied. Four key themes were identified regarding the contribution of WIL to curriculum and student learning: (1) the relationship between WIL and the program curriculum; (2) WIL as a learning process; (3) learning from diverse WIL contexts; and (4) developing competence through WIL. Overall, the diversity in the standards reflected differing understandings of what WIL is. Thus, in the absence of frameworks for designing accreditation standards, the risk is that some AH professions will continue to perpetuate the myth that the primary purpose of WIL is to provide a bridge between theory and practice.

## 1. Introduction

The global demand for allied health (AH) professionals is increasing in response to diverse challenges, including complex and multiple chronic health conditions, aging populations, the impact of COVID-19, and the changing expectations of service users or clients, carers and communities [1,2]. With this comes increased public expectations for AH professionals to be safe and competent practitioners [3], leading professional and regulatory bodies to transition from broad practice principles to professional standards or competencies [4]. These professional standards or competencies inform the professional bodies’/regulatory authorities’ curriculum standards against which each AH professions education program is reviewed and accredited. On successful completion of their degrees, AH professionals are deemed to be able to practice as semi-autonomous or autonomous practitioners. To support this capability, a core component of AH professional degrees is work-integrated learning (WIL), also known as clinical education, fieldwork placement or practicum placements.

### 1.1. Definitions of Work Integrated Learning

Whilst multiple definitions for WIL exist [5], the term WIL is often used as an umbrella term for learning experiences where students, as part of a purposefully developed curriculum, integrate theory with the practice of work [6]. Further, the Tertiary Education Quality and Standards Agency states that WIL includes any “arrangement where students undertake learning in a work context as part of their course requirements” [7] (p. 1), that is, “WIL describes strategies and activities that promote student learning through engaging with aspects of work” [8] (p. 12). Thus, WIL is a pedagogical approach that promotes student learning and prepares graduates for their professions [8]. WIL encompasses a range of activities including professional workplace placements, as well as online or virtual and/or simulated work environments. The extent and format of the WIL aspect of the curriculum varies in structure, models [9], supervisory arrangements, length and type of placement, and timing within the program [6,7,9,10,11]. The value of WIL in supporting students’ progression towards being a work-ready graduate is identified by the professions [11,12], the regulatory bodies (see for example [10]), the tertiary institutions that deliver the programs [13,14], and students [15,16]. However, despite the acknowledged importance of placements, a common language and conceptual understanding of the pedagogic processes underpinning WIL is lacking [17,18].

### 1.2. The Accreditation and Benefits of WIL

The benefits of WIL for students include the development of professional competencies required for practice and, through the process of professional socialization [13], the development of the student’s professional identity [15,19]. WIL also helps students to develop general professional skills such as effective communication and teamwork [14], in addition to discipline-specific technical skills. This occurs through the application of theoretical knowledge first introduced in the university context [11,14,20]. Boud et al. [21] argue that authentic workplace learning experiences support students’ capacity for problem-solving, decision-making and reflexivity. These capacities are key components of work-readiness, as described in the literature [22], and underpin attributes that are highly sought after by employers, such as personal insight and self-awareness, resilience, professionalism and a commitment to lifelong learning, along with communication and organisational skills [23,24].

Health professional regulatory or professional bodies ensure that AH education programs provide sufficient and quality WIL opportunities, as outlined in the discipline-specific professional competencies and educational accreditation standards. Such bodies regularly monitor education providers’ adherence to said standards, ensuring that successful graduates from that program can be considered safe and competent practitioners [11] and thus eligible to practice their profession as registered or accredited health professionals. The educational accreditation standards outline what is to be learned and how [25], with tertiary providers using these standards to design and implement their curricula. Thus, by the very nature and wording of the standards, each profession’s regulatory authority inherently articulates, defines and describes the criteria for what constitutes “work integrated learning” in their profession. That is, while the accreditation standards may stipulate lengths of placement, hours, appropriate contexts and even supervisory models, we propose that inherent in all of these practical requirements is an implicit understanding or conceptualisation of this learning experience known to all in the profession as “a placement”.

### 1.3. Our Research

One of the only studies to consider the accreditation requirements for placements across a range of AH disciplines is that by McAllister and Nagarajan [26]. These authors found little comparability across six Australian AH professions, citing differences in hours of clinical practice (from no specified hours for professions such as physiotherapy and speech pathology, to 500 h for exercise physiology or 1000 h for occupational therapy), placement contexts or types of conditions students should experience, supervisory requirements, and supervisor ratios. This study was undertaken in 2015, based on documents that were published between 2006 and 2014, and was limited to occupational therapy, speech pathology, exercise physiology, physiotherapy, rehabilitation counselling and medical radiation practice. Accreditation documents are usually updated every 5–10 years; therefore, the information published in their article regarding WIL may have changed. In addition, compared with McAllister and Nagarajan’s [26] study, we aimed to dive deeper and across a broader range of professions to understand how different Australian AH professions conceptualise WIL.

The aim of this study was to identify and compare the education standards for WIL across AH professions in Australia and, through this process, to explore the concept of “work integrated learning”, as articulated in AH professions’ accreditation standards. Two research questions guided the study:

RQ1: What are the similarities and differences between Australian AH professions in their WIL requirements?

RQ2: How do Australian AH professions conceptualise WIL as evidenced in their accreditation requirements?

## 2. Methods

The design of this study was underpinned by both positivist [27] and constructivist [1,28] research paradigms. To answer the research questions, a non-experimental explanatory mixed-methods research design [29] consisting of two stages was selected to guide the collection and analysis of data obtained from publicly available AH professions documents. Stage one proposes that the accreditation documents are the “source of truth”, and therefore our answers to Research Question 1 could be extracted from these documents (positivist approach [27]) to collate and compare data. Stage 2 of the project uses a constructivist approach [1,28] to analyse and interpret the conceptual understandings of WIL, as articulated in the documents [29]. This approach sought to understand and interpret the data to elicit the breadth of conceptual constructs across the professions, rather than compare the findings (i.e., Research Question 2).

This duality of positioning is also a reflection of the research team involved. As recommended by Olmos-Vega et al. [30] for health profession education researchers, we next outline the professional backgrounds of each author, which may have shaped decisions made throughout the project. Two of the authors (MP, YT) are registered occupational therapists and academics with expertise in WIL and a third co-author (JR) is an academic with expertise in exercise physiology curriculum and WIL. All have extensive experience of teaching and leadership roles within their profession and across allied health. These three authors have been extensively involved in or supported the accreditation processes for their respective professions. The other three co-authors (AK, RYRL, KS) bought to the research process their previous roles in medical student journal editing, systematic review and meta-analyses, data entry and analysis and a relative distance and naivety from AH accreditation that allowed fresh and novel perspectives. In addition, the team members engaged in an ongoing process of collaborative interpersonal reflexivity to further explore the interplay between motivations, assumptions and expectations [30] across the team and how these influenced decisions made throughout the research process.

Document analysis was used to identify, compare and explore WIL across AH professions in Australia [31]. Document analysis is a systematic process involving selecting documents, reviewing their contents, extracting information, and interpreting and synthesising data (excerpts, summaries, quotes, etc.) that relate to the research questions [31]. We considered the analysis of health professional documents, specifically regulatory authority/professional bodies that published accreditation standards or guidelines and supporting documents, to be an appropriate methodology as these documents inform education-providers of the purpose, requirements and expectations of WIL and would provide insight into how different professional groups conceptualise WIL.

### 2.1. Allied Health Professions Inclusion/Exclusion Criteria

A list of potential AH professions that could be included in the document analysis was derived from the work of Turnbull et al. [32], and the Allied Health Professions Australia [29] website. From this list, professions were included if:The education program is provided by an Australian university, at Australian Qualifications Framework (AQF) Level 7 or higher, and is eligible for accreditation by the relevant national accreditation body;The program curriculum includes a WIL component that involves significant service user/client/patient interaction;The AH practitioner has a direct patient care and treatment role;The profession has robust and enforceable regulatory mechanisms;The profession has clearly articulated national competency standards;The profession has a national professional organisation with a code of ethics/conduct and clearly defined membership requirements.

AH professions whereby graduates are not fully accredited/registered upon completion of the qualifying degree, for example, those professions where an internship year is required, were excluded. Non-AH professions, for example nursing, medicine and dentistry, were also excluded.

### 2.2. Document Analysis

This study drew from the document analysis process outlined by Altheide and Schneider [33]. Four broad stages of analysis were conducted: becoming familiar with the documents; protocol development and data collection; data analysis; and reporting.

Initially, the research team identified the goal of the project and the sources of relevant information, familiarised themselves with the range of sources and selected the units of analysis. These “units”, which were the documents relating to professional and accreditation standards, including support guides, were retrieved from websites and via email request from the various AH professions included in the study. The retrieved documents represented those that were current as of February 2023. To support a consistent approach to data collection, the research team developed a draft protocol of the categories of information that would be extracted from the documents. The protocol was tested using a subset of documents and further refined in an iterative process until a final protocol was agreed upon by the research team. The data were then extracted from all documents and organised in a tabulated format in March–April 2023. All documents were reviewed by two team members to ensure the completeness and validity of the data table.

### 2.3. Data Analysis

Objective data (i.e., length of program and length of placements) were collated and tabulated for comparison. Textual data (i.e., supervision requirements and assessment processes) were organised into categories, using an inductive approach. The data were analysed to compare, combine and critically interpret the range of similarities and differences in WIL across the AH professions [31].

A thematic analysis was conducted to understand the underlying philosophy of WIL as outlined in the documents [34]. Our aim was to find out how the accreditation documents expressed or articulated the fundamental nature of WIL in the program design and delivery. All data from the data extraction table relating to how WIL was situated in the overall curriculum were copied and pasted verbatim into a separate document. The original documents were checked again to ensure the key text was included. The text was read several times and then coded [35]; key phrases and concepts were identified and highlighted by YT and checked by JR. Once agreement was reached, the codes were then collated into concepts by YT and discussed with JR and MP until consensus was achieved, with the concepts then grouped into larger themes.

## 3. Results

### 3.1. Allied Health Professions

From an initial list of 27 AH professions, the research team identified 17 that met the inclusion criteria. Diabetes educators were initially considered for inclusion; however, further exploration of the education requirements indicated this profession did not meet the inclusion criteria due to an insignificant component of WIL. It is noteworthy that diabetes educators are also frequently registered health professionals before becoming accredited educators. Orthoptics was subsequently excluded from the study as attempts to obtain documents form the Australian Orthoptic Board were unsuccessful. Music therapy was also excluded as the available documents did not provide sufficient data for extraction.

Of the 15 professions remaining in the study, nine are part of the Australian Health Professional Registration Authority (AHPRA) and six are part the National Alliance of Self-Regulating Health Professions (NASRHP). These national organisations are designed to provide public assurance about the training, quality and safe practice of health professionals. The documents consulted as part of this study are shown in Table 1. We observed a consistent approach in the presentation of the accreditation standards of the AHPRA health professions. That is, all but one of these professions organised their standards into the following five general categories: public safety; academic governance and quality; program of study; the student experience; and assessment. While also using these same categories, optometry includes a sixth category: cultural safety. The presentation of the accreditation standards of the NASRHP professions was individualised to each profession. We also found the AHPRA professions to be somewhat similar in how they presented their WIL expectations. One exception was occupational therapy, where additional information is included as part of the World Federation of Occupational Therapy (WFOT) expectations. In contrast, the NASRHP professions, while overlapping to some extent with the AHPRA professions, include additional information. For example, exercise physiology says that practicum allows for students to gain interprofessional experience, and social work says that field education allows for socialisation into the profession and for students to build a professional identity. Speech pathology emphasises the developmental trajectory of students towards graduate-level practice.

### 3.2. Terms and Descriptions

The terms used to describe WIL and the expected outcomes of WIL are presented in Table 2.

There is no consistent term for WIL among the 15 allied health professions, with some professions using multiple terms throughout their documents. Descriptions and definitions of the terms add to the diversity of understandings across AH professions, and therefore were grouped together for comparison and meaning.

“Work integrated learning” was used in the standards of four professions: Chinese medicine, paramedicine, medical radiation practice and podiatry. Dietetics used the closely related term workplace learning, while also using professional placements. Paramedicine makes reference to the TEQSA definition of work-integrated learning: “…any arrangement where students undertake learning in a work context as part of their course requirements” [7] (p. 1), although it is noted that paramedicine specifically states that the learning occurs in a workplace external to the education institution. This discrepancy may be due to a recent update of the TEQSA Guidance Note. Chinese medicine, podiatry and medical radiation practice refer to work-integrated learning as being an “umbrella” term, which includes a range of approaches and environments that permit the integration of theory with practice. Furthermore, the environments can be both internal and external to the education institution and the experiences can be real or simulated. The term work-integrated learning seems to emphasise the location and context of the student experience.Terms such as “clinical placement”, “clinical experience”, and “clinical education” are used in chiropractic, optometry, orthotics and prosthetics, osteopathy, paramedicine, physiotherapy and medical radiation practice. The use of “clinical” as a descriptor of WIL appears to be aligned with the nature of the work the student is engaging in and not a descriptor of the workplace context. Similar to the use of the term WIL, there is an emphasis on clinical placements (or similar) providing opportunities to integrate theory into practice in a professional environment.Terms such as “placement”, “professional placement”, “practicum”, “practice placement” are used in dietetics, exercise physiology, osteopathy, paramedicine, and occupational therapy. Exercise physiology considers practicum as experience in a real-world context at a placement site. Both exercise physiology and osteopathy state that placement provides a context for the application of theoretical knowledge in practice. Osteopathy specifically excludes simulation but is inclusive of a range of external health and care environments.“Field education” is the preferred term in rehabilitation counselling and social work and “fieldwork” is mentioned in occupational therapy. There is no consistent definition of the term, although rehabilitation counselling described field education as experiences in industry-based settings that allow for supervised activities to be undertaken.Practice education is the term used by speech pathology, and occupational therapy. In speech pathology, practice education includes simulated learning, university managed clinics and external workplaces, suggesting the focus is on its educational purpose rather than the location.

The diversity of terms used and definitions of WIL in AH suggest there are three or more factors that are considered: the location or context of the learning activity, the nature of the work activities being performed by the student, and the focus of the learning that is taking place, with an emphasis on integrating theory with practice.

### 3.3. Expected Outcomes/Requirements

All professions except rehabilitation counselling specify competence as an outcome of WIL (Table 2). This is often represented in the form of a specific standard such as “The quality and quantity of clinical experience are sufficient for developing a student to be a graduate competent to practice” [37] (p. 7), but similar standards appear in many of the accreditation documents (see Appendix A). While most of the professions require only competence as the outcome, five also include a time component, ranging from 200 h in any year for rehabilitation counselling and 360 h in exercise physiology to 100 days in dietetics and 1000 h in social work and occupational therapy.

Most accreditation documents include a requirement for diversity of experience (see Appendix A), suggesting that students should have more than one placement and work with patients/clients with different needs during the program. Some professions explicitly state if simulated placements are acceptable. Some professions provide explicit requirements related to direct service provision (for example, rehabilitation counselling and social work), the volume of overseas placement (for example, social work and speech pathology) and the specific area of practice being experienced (for example, exercise physiology).

### 3.4. Supervisor and Supervision Requirements

Most accreditation documents state that supervisors must be suitably qualified and experienced, and/or hold registration or accreditation (see Appendix A). Supervision by other health professionals is explicitly permitted by chiropractic, exercise physiology, occupational therapy, optometry, osteopathy, physiotherapy and podiatry. Of the other professions, only orthotics and prosthetics, medical radiation practice and social work explicitly state that supervision is to be provided by those within the profession. Rehabilitation counselling, exercise physiology and social work are the only professions to stipulate a supervisor-to-student ratio for placement supervision, although osteopathy requires the ratio to be “appropriate”. Rehabilitation counselling and social work are the only professions to stipulate a minimum number of hours of formal supervision per week.

Relatively few standards explicitly require the supervisor to be trained in supervision or have supervisory skills. These professions include optometry, occupational therapy, and orthotics and prosthetics. The role of the supervisor and of supervision is frequently vague, and generally includes either responsibility for or contribution to student assessment (see Appendix A).

### 3.5. Assessment Requirements

The most frequent requirement for assessment is that student performance is assessed against the professional competencies or standards for practice (see Appendix A). No professions require a specific performance assessment tool to be used when assessing students on placement, although three professions have an assessment tool that is widely used: occupational therapy, physiotherapy and speech pathology. The assessment tools are referred to in the occupational therapy and speech pathology accreditation documents.

Most professions indicated that a choice of assessment tools and modalities was applicable. Valid and reliable assessment was emphasised across the AHPRA profession, although that is not to suggest that this is also not important for the NASRHP professions.

### 3.6. Conceptualisation of WIL in AHP Accreditation Standards

Thematic analysis of the information pertaining to the conceptualisation of WIL, as a component of curriculum, within the accreditation documents across the AH professions, resulted in four themes: (1) the relationship of WIL to the program curriculum; (2) WIL as a learning process; (3) learning from diverse WIL contexts; and (4) developing competence through WIL.

### 3.7. Theme 1: The Relationship of Work-Integrated Learning to the Program Curriculum

While most accreditation documents included WIL in the program structure and delivery, not all required it to be explicitly included as a component of the program curriculum. The occupational therapy program accreditation document in Australia, refers to the WFOT standards, which explicitly refer to the central place of WIL within the curriculum and education: “Practice Education is central to the educational process. It includes curriculum content and is an education method, but is presented in a separate category because additional standards apply” [56] (p. 48).

Other accreditation documents highlighted the importance of integration between program curriculum and WIL, for example, the Standards for Orthotic/Prosthetic Tertiary Education Programs in Australia [60] state that: “Clinical placements are integrated into the curriculum to allow the achievement of competencies across a range of practice areas relevant to the Australian context” [60] (p. 15).

The integration of WIL within the curriculum suggests an understanding that WIL is separate from, but consistent with, the program curriculum and educational methods. For example, Chinese medicine and paramedicine use the following clause: “The curriculum design includes vertical and horizontal integration of theoretical concepts and practical application throughout the program, including simulation and work-integrated learning experiences” [37] (p. 19).

More specific direction is provided by Speech Pathology Australia (SPA), requiring programs to provide a description of the philosophy and pedagogy that inform the academic and WIL aspects of the program, stating that: “Universities must submit an outline of the pedagogical framework that underpins their practice education program, and how it integrates with the overall academic program” [79] (p. 8). Conversely, the accreditation standards for social work field education state that “Field education is a distinctive pedagogy for social work education” [77] (p. 2).

While most standard documents did not refer to an underlying philosophical or pedagogical foundation for curriculum, several accreditation documents placed the onus on the curriculum developers to articulate the link between the on-campus learning and WIL. For example, the Australian Physiotherapy Council (APC) requires that an educational philosophy informs the program of study design and delivery and includes the “sequencing of units of instruction and clinical placements” [69] (p. 10).

Many documents referred to the need for preparatory or pre-requisite learning prior to the placement of students in the practice environment. For some, the preparatory learning focused on knowledge and skills (for instance, common pathologies, or unspecified practical skills). In other documents, preparation included demonstration of professional behaviours and capabilities (including fitness to practice, safe patient care and infection control requirements). “Students in the program are required to achieve relevant pre-clinical capabilities before each period of work-integrated learning” [37] (p. 8). The requirement for pre-requisite learning and adequate preparations may arguably indicate the integration of course content and WIL within the curriculum.

### 3.8. Theme 2: Work-Integrated Learning as a Learning Process

A common concept articulated within the standards is that WIL involves the student’s active engagement in integrating the theoretical content taught in the course and practical experience. The student’s active engagement in integrating theoretical knowledge implies that a learning process is anticipated. In most of the documents that referred to this concept, the integration was of theory with practice, rather than of theory into practice. For example, paramedicine defines the term clinical placement as “A term used for a range of approaches and strategies that integrate theory with the practice of work within a purposefully designed curriculum” [66] (p. 25), while optometry defines clinical placement as providing opportunities for students “for the purposes of integrating theory into practice” [58] (p. 25). The implicit relationship between theory and practice is fundamentally different; however, the outcome, as a learning process for students, may be the same.

A second related concept indicating student learning through WIL is the application of knowledge, skills and, sometimes, professional attributes. For example, Chinese medicine and podiatry are similarly written, with the podiatry standards including a statement that WIL will “produce a graduate who has demonstrated the knowledge, skills and professional attributes to safely and competently practise” [72] (p. 19). Likewise, the definition of clinical placements in the optometry standards includes “building the knowledge, skills and attributes essential for professional practice” [58] (p. 25). While the way in which the student will learn through the application of knowledge and skills is not highlighted, it is recognised that this will enable the student to demonstrate professional competence.

The two concepts of the integration of theory and practice and application of knowledge, skills and (less frequently) attributes suggest that WIL involves students going beyond professional knowledge, (declarative knowledge) and learning how to use that knowledge (procedural knowledge) in a diverse range of practice situations.

The Australian Social Work accreditation standards include an explicit expectation that WIL requires student to “reflect on and refine their ways of thinking, doing and being” [77] (p. 2). In this standard, WIL goes beyond the application of knowledge and skills; rather, it is viewed as “a constructive and reciprocal learning space to develop” [77] (p. 2) as a health professional, with the goal that “Students make sense of what it means to be a social worker by developing their professional identity, integrity and practice frameworks” [77] (p. 2).

The accreditation standards for exercise physiology state that both practical learning and knowledge translation should be included in the course design so that WIL placements provide more effective learning experiences for students. “The course design should also ensure student placement can be a meaningful experience by undertaking essential units of study to translate knowledge and skills, particularly those practical in nature” [45] (p. 7). The above statement suggests that the translation of knowledge and skills is not peculiar to WIL and can be facilitated by a good course design. However, student learning during WIL involves learning how to become competent and confident in using their knowledge and skills as a health professional in a practice context. As described in the Australian Social Work accreditation standards, “field education aims to provide students with a robust and fulfilling learning experience from which they gain a strong sense of professional competence and the confidence that they are ready to enter the profession” [77] (p. 2).

### 3.9. Theme 3: Learning from Diverse Work-Integrated Learning Contexts

As outlined in Appendix A, there is broad agreement that WIL can occur in a variety of settings, locations, and contexts, explicitly including simulation in some professions. “Work-integrated learning can include clinical practice, community education programs, and laboratory work (such as orthoses manufacture) and it can be done in person or in a range of simulated learning environments” [72] (p. 9). Similarly, paramedicine states WIL can take place in “clinical or other professional placements, online projects, internships, or workplace projects” [66] (p. 9) and “recognises that some clinical placements may take place in facilities that are not health services or health facilities and are not required to be accredited or licensed” [66] (p. 10).

The breadth of practice contexts is somewhat explained by four professions (dietetics, paramedicine, speech pathology and occupational therapy) who articulate the changing and emergent nature of practice, highlighting the need for students to be equipped for new practice contexts on graduation. The requirement for diversity is not limited to the context for practice but includes the requirement to work with different populations, with different needs and circumstances. “Students are provided with practice education experiences with individuals and communities across the lifespan in a range of contexts and with a range of populations” [80] (p. 30). Breadth of experience is encouraged to compensate for the impossibility of students experiencing every possible kind of practice setting. In this way, there is an implied understanding that students will learn from the context in which the WIL experience occurs, and that a diversity of contexts will enable students to apply their knowledge and skills in different ways, and most effectively prepare for a changing professional practice. “Each student is provided with a variety of workplace learning experiences reflecting socio-ecological approaches to health, major health priorities and the broad landscape of dietetic practice, including policy and the provision of services and care to individuals, groups, communities and populations, which allows them to meet the NCS” [43] (pp. 16–17).

Some standard documents refer to the idea that WIL provides an opportunity for engaging with the professional industry, or the real world of practice, so that students become familiar with the professional environment into which they will graduate. Specifically, the Exercise & Sport Science Australia and Speech Pathology Australia documents mention the inclusion of interprofessional practice skills as being achieved in practice settings. “Practicum provides opportunities for students to engage with industry, undertake workplace tasks and gain experience in inter-professional practice” [48] (p. 2).

### 3.10. Theme 4: Developing Competence through Work Integrated Learning

The development and achievement of professional competence was commonly referred to as an outcome of WIL. Many standards explicitly referred to the professional competency standards for registered practitioners, although some noted that the level of expectation would be considered entry-level. Progression towards these competency outcomes was not determined in the standards, which required program documentation to articulate the expectations of student competency in the learning outcomes with valid and reliable assessments of WIL placements. An alternative approach is used in the Dietitians Australia standards, where a programmatic assessment was used, suggesting that the achievement of graduate competencies could be achieved flexibly across the program. In whatever way competence is assessed, there is an implied expectation that WIL will provide opportunities to learn and demonstrate professional competencies. While the assessment of competence does not solely rely on performance in WIL, competency-based assessment by a registered health professional is frequently a requirement of accreditation.

Conversely, two accreditation standard documents referred to a holistic view of competency, which included students’ capacity to provide a service and the development of professional identity. While competency assessments are generally/traditionally achieved through observable actions and demonstration of skills, these more holistic goals suggest that WIL outcomes include both the creative and personal learning required to become a professional. “Field education […] enables students to integrate classroom learning with professional practice so that students reflect on and refine their ways of thinking, doing and being” [77] (p. 2).

## 4. Discussion

Although allied health professions are used as an overarching term for those professionals providing a range of services to support Australians health and well-being [32], each have their own distinct professional identity, which is reflected in their professional competency standards [3], and thus their education accreditation standards [81]. Similarly, to McAllister and Nagarajan’s earlier investigation of six AH professions’ accreditation standards (dated 2006–2014) [26], we also found marked differences in the quantity, volume and mechanisms for the assessment of student learning and supervisory requirements. In addition, our analysis also included the terminology used for WIL, as well as the outcomes of this critical component of the degree (i.e., competence and/or minimum required placement hours), with variations in these factors across the different professions.

To our knowledge, this is the first Australian-based study that has aimed to understand how WIL is conceptualised by AH professions through the wording of their accreditation standards. Our analysis of the intent of the standards suggested that, across the 15 AH professions, there was diversity in how integral and core to the curriculum WIL is considered to be. Diversity also existed in the importance given to explicitly articulating the pedagogical underpinnings of the placement components (the relationship of WIL to the program curriculum); the ways in which learning is conceptualised, that is, the process through which student learning is achieved in the practice context (WIL as a learning process); the contexts that a student must experience (learning from diverse WIL contexts); and how competence in WIL occurs (developing competence through WIL).

It is interesting to note the processes by which the accreditation standards may have been established. Very few of these standards appear to be evidence-based [10], with many appearing to be informed by tradition [82] or expectations of the profession or employers, [26] or determined based on the expertise, perspectives or mandates of the authors of the standards, working with existing resources and/or timeframes [83]. Furthermore, in some cases, the actual wording of the standards appears to be carried over from other disciplines. For example, the occupational therapy accreditation standard authors argue for the continued 1000 h requirement (as well as competency), stating that these hours have been consistent in the history of the profession and are also comparable with other health profession educational programs [56]. A similar approach has been observed in the development of professional competency standards. For example, Allen and Palermo [84] describe a process of revising the wording of some of their dietetic competency standards to be consistent with other profession’s competency standards.

As noted by others [81,85], there appears to be little guidance for those developing competency standards or frameworks [84], with Shaw and Tudor [84] suggesting that some regulatory authorities may lack the educational expertise and access to the evidence-base required to develop accreditation standards. In this context, frameworks that assist regulators in developing both their competency and accreditation standards are required. This need is addressed by Batt et al. [86], who published a six-step framework for developing competencies for health professions. Batt et al. [86] argue for the importance of a rigorous process for developing competencies, as these inform the downstream processes such as the design of education accreditation standards. Frameworks for the design of accreditation standards have been published for medical education [87], but no specific framework or process could be found for AH professions. Indeed, we argue that, based on our analysis, it is not only important to be able to present the process by which the educational accreditation standards have been designed, but that, within this process, the designers also need to have articulated their conceptualisation of key curriculum components, such as WIL. Just as Batt et al. argues that “how a profession conceptualizes competence (e.g., degree of granularity from atomistic to holistic/integrated) will influence how developers decide to represent competence in a framework” [86] (p. 931), we too propose that how developers conceptualise WIL is how it is then represented in the accreditation standards.

It became apparent through the analysis that in just over half of the AH professions’ accreditation standards, WIL was understood to be the “place” where students practice how to apply the “theory” learned in the classroom. This notion of “theory to practice” is a common theme throughout AH education, for example, providing opportunities for students to “apply[ing] theory to practice” [88] (p. 720) or to “link theoretical learning to the application of authentic work-focused tasks, requirements and practices” [89] (p. 34), or that placements exist to “bridg[ing] the gap between university classroom and professional practice” [90] (p. 327). However, placements appear to offer more than just being a bridge between two worlds. Recent studies [16,91] of early-year placements in occupational therapy have shown that students perceive that they learn not only about the practices of the profession but about themselves and the service user journey. Perspectives of WIL amongst other professions also demonstrate a range of learning that occurs through placement. In teacher education, WIL is observed to develop employability skills such as teamwork, communication and problem-solving, and to provide a foundation for lifelong learning [8]. Similarly, Winchester-Seeto and Piggott [92] propose a focus on ‘work-force’ learning, rather than ‘work-place’ learning, with ‘work-force’ learning including a broader range of settings and learning activities such as virtual placements and simulation to enhance students’ critical thinking and ability to articulate professional identity. As the nature of work continues to change, these capabilities will more effectively prepare graduates for professional practice in the future [8,92].

Patton et al. [93] describe placements metaphorically as crucibles for student learning, a space that is rich in learning experiences arising out of the interaction between influences in the workplace, professional practice engagements, the actions and intentions of clinical education, and the students’ experiences and dispositions. Thus, to describe the purpose of the placement components of the curriculum as being primarily a space where students learn to bridge theory to practice is to perpetuate the myth that theoretical learning occurs in the classroom and practical learning in the placement environment. In continuing to use this description, Björck [20] argues that we create a dualism that students also adopt; however, by moving towards a non-dualistic meaning, graduates could transition into the workforce understanding how to use both research-based and practice theories in an integrated manner, with this being the practical wisdom Lawton et al. [22] considers as one of the constructs of work readiness. While it is outside the scope of this study to speculate on how each profession conceptualises placements at a deeper level than the descriptions provided in the accreditation standards, further research is needed to clearly articulate “what is a placement”, what students learn through this experience, and in what ways this learning contributes to the development of the work-ready graduate. These understandings can then inform the development or ongoing review of the profession’s accreditation standards.

## 5. Limitations

This study explored the similarities and differences in how Australian AH professions conceptualised WIL, as evidenced through their accreditation requirements. The limitations that may impact our findings are similar to those discussed in Bogossian and Craven’s [85] study of the presence of interprofessional education in accreditation standards. That is, there is no one definitive list of what is and is not an AH profession. Thus, our sampling process may have excluded some unregulated or self-regulating professions. In addition, our document analysis only included publicly available program accreditation standards. Our aim was to gain an in-depth understanding of how WIL was conceptualised across the professions through an analysis of the wording of the standards. Whilst every effort was made to ensure that our interpretations were grounded in the descriptions of the standards, we also acknowledge that what we have presented may or may not match the actual understandings of a profession. Finally, as this study specifically focused on the conceptualisation of WIL within the accreditation standards for each profession, the findings do not represent the various conceptualizations of WIL that may have been drawn from the written curriculums of professional education programs or more directly from curriculum developers, practicing professionals, or the students undertaking WIL as part of their education. Further research into the perspectives of other stakeholders might reveal very different conceptualisations of the purpose, value and learning outcomes of WIL.

## 6. Conclusions

The accreditation of educational programs is growing apace as AH professionals recognise the need to assure the public that those graduating from educational programs are safe and competent practitioners, with employers seeking work-ready graduates. Whilst the whole degree contributes to the readiness of the AH professional to practice semi-autonomously or autonomously, the findings of this study demonstrate that the WIL components of the degree contribute significantly to the student’s progress towards being that work-ready graduate. The importance of this is highlighted in the education accreditation standards, with sections often “set-aside” with specific standards related to WIL. This study aimed to understand the similarities and differences between Australian AH professions in terms of their WIL requirements, and found multiple terms and definitions of WIL exist, with inconsistent requirements for the amount and type of settings. This finding suggests that it may be challenging for those who develop and/or review their profession’s accreditation documentation to develop valid standards. Whilst frameworks for the development of competencies exist, these do not appear to be available for accreditation standards and the process by which accreditation standards are developed may only be included in the accreditation documentation, and thus not subject to critical peer review.

The study also aimed to discover how Australian AH professions conceptualise WIL in their accreditation requirements. Across the 15 AH professions, we found diverse understandings of WIL, leading us to question how WIL is being conceptualised. At the simplest level, WIL is described as being the bridge between theory and practice. However, we argue that to only consider WIL in this way is to do a disservice to what WIL actually offers. As a crucible for student learning, WIL provides rich learning experiences. However, to fully understand what these are, each AH profession needs to consider how they conceptualise placements and how this is then articulated through their accreditation standards. Further research is needed.

## Figures and Tables

**Table 1 ijerph-20-06478-t001:** Allied health professions and documents reviewed.

Allied Health Profession	Accreditation Body/Accreditation Authority	Documents Consulted
Chinese Medicine *	Chinese Medicine Accreditation Committee	1. Professional Capabilities for Chinese medicine practitioners (2019) [36]
		2. Accreditation standards: Chinese medicine (2019) [37]
		3. Guidelines for accreditation of education and training programs (2020) [38]
Chiropractic *	Council on Chiropractic Education Australasia	1. Accreditation Standards for Chiropractic Programs and Competency Standards for Graduating Chiropractors (2017) [39]
		2. Accreditation Guidelines for Chiropractic Education Programs (2018) [40]
		3. Accreditation Policies & Procedures (2015) [41]
Dietetics #	Dietitians Australia	1. Accreditation Standards for Dietetics Education Programs Version 3.0 (2022) [42]
		2. Evidence Guide for Accreditation of Dietetics Education Programs: Examples of Evidence (2022) [43]
		3. Accredited Practising Dietitian (APD) Policy 2016 [44]
Exercise Physiology #	Exercise & Sports Science Australia	1. Course Accreditation Guide Version 6 (2022) [45]
		2. Accredited Exercise Physiologist Professional Standards (2021) [46]
		3. Accredited Exercise Physiologist Professional Standards Support Guide (2020) [47]
		4. Practicum Standards (2022) [48]
		5. Course Accreditation Requirements (2021) [49]
		6. Course Accreditation Standards (2022) [50]
Medical Radiation Practice (includes diagnostic radiographer, nuclear medicine technologist and radiation therapist)	Medical Radiation Practice Accreditation Committee	1. Accreditation standards: Medical radiation practice (2019) [51]
		2. Professional capabilities for medical radiation practitioners (2020) [52]
Occupational Therapy *	Occupational Therapy Council of Australia Ltd.	1. Accreditation Standards for Australian Entry-Level Occupational Therapy Education programs (2018) [53]
		2. Guidelines and Evidence Guide for the accreditation of Australian entry-level occupational therapy education programs (2022) [54]
		3. Occupational Therapy Council Accreditation Standards. Explanatory Guide: the use of simulation in practice education/fieldwork (2020) [55]
		4. World Federation of Occupational Therapists Minimum Standards for the Education of Occupational Therapists (2016) [56]
		5. Australian occupational therapy competency standards (2018) [57]
Optometry *	Optometry Council of Australia and New Zealand	1. Accreditation standards and Evidence Guide for Entry-Level Optometry Programs (2023) [58]
		2. Entry-level Competency Standards for Optometry (2022) [59]
Orthotics & Prosthetics #	Australian Orthotic Prosthetic Association	1. Course Accreditation Standards for Orthotic/Prosthetic Tertiary Education Programs in Australia (2020) [60]
		2. Entry Level Competency Standards for Australian Orthotist/Prosthetists (2014) [61]
		3. Competency Standards wall chart [62]
Osteopathy *	Australasian Osteopathic Accreditation Council	1. Osteopathic Accreditation Standards (2021) [63]
		2. Osteopathic Accreditation Standards Essential Evidence (2021) [64]
		3. Capabilities for Osteopathic Practice (2019) [65]
Paramedicine *	Paramedicine Accreditation Committee	1. Accreditation standards: Paramedicine (2020) [66]
		2. Professional capabilities for registered paramedics (2021) [67]
Physiotherapy *	Australian Physiotherapy Council	1. Accreditation Standard for entry-level physiotherapy practitioner programs (2016) [68]
		2. Guidelines for accreditation, entry-level physiotherapy practitioner programs of study (2021) [69]
		3. Physiotherapy practice thresholds in Australia and Aotearoa New Zealand (2015) [70]
		4.Australian Physiotherapy Council (2021) [71]
Podiatry *	Podiatry Accreditation Committee	1. Accreditation standards: Entry-level podiatry programs (2021) [72]
		2. Accreditation standards: Summary (2021) [73]
		3. Professional capabilities for podiatrists (2022) [74]
Rehabilitation Counselling #	Rehabilitation Counselling Association of Australia	1. Accreditation Manual for Rehabilitation Counselling Education Programs (not dated) [75]
Social Work #	Australian Association of Social Workers	1. Australian Social Work Education and Accreditation Standards v2.1 (2021) [76]
		2. ASWEAS Field Education Standards (2021) [77]
Speech Pathology #	Speech Pathology Australia	1. Guidelines for the accreditation of speech pathology degree programs, Part 1: Regulations, standards and procedures (2022) [78]
		2. Guidelines for accreditation of speech pathology degree programs, Part 2: Evidence Guide (2022) [79]
		3. Guidelines for the accreditation of speech pathology degree programs, Excerpt: Accreditation standards and criteria (2022) [80]

* AHPRA profession; # NASRHP.

**Table 2 ijerph-20-06478-t002:** Terminology and expected outcomes of work-integrated learning.

Allied Health Profession	The Term(s) Used to Describe Work Integrated Learning	Expected Outcomes/Requirements
Chinese Medicine	Work-integrated learning	Competence
Chiropractic	Clinical experience placements; clinical placements; clinical experiences	Competence
Dietetics	Workplace learning/workplace learning experiences; professional placement; placement program	Competence and a minimum of 100 days
Exercise Physiology	Practicum, student placement, placement, practicum placements	Competence and a minimum of 360 h in a variety of activities to demonstrate attainment of competency in exercise assessment and prescription and delivery. Of these hours:
		- At least 200 h across the Accredited Exercise Physiology (AEP) core areas of practice
		- Remaining 160 h can be in any area across the AEP scope of practice, but with no more than 100 h across the emerging or niche areas of practice
Occupational Therapy	practice education; fieldwork; practice placements	Competence and 1000 h *
Optometry	placements; clinical placements; clinical training; clinical experiences	Competence
Orthotics & Prosthetics	clinical placement	Competence
Osteopathy	clinical placement; professional placement; clinical experiences	Competence
Paramedicine	clinical placements; professional practice experience; work-integrated learning	Competence
Physiotherapy	clinical education; clinical placement; clinical training	Competence
Podiatry	work-integrated learning	Competence
Medical Radiation Practice	work-integrated learning; clinical placement	Competence
Rehabilitation Counselling	field education	A minimum of 200 h field education in any year, which includes a minimum of 80 h in direct service provision and direct client contact
Social Work	field education; field placement	Competence and:
		Minimum of 1000 h across no more than three placements with none shorter than 250 h.
		At least two of the placements require distinctly different experiences.
		At least 500 h must be undertaken in Australia.
		Of the 1000 h, 500 h must be in a direct practice role.
Speech Pathology	Practice education; placement	Competence

* 1000 h is the WFOT international benchmark.

## Data Availability

The included papers are referenced within, and data extraction tables are included within and in Appendix A.

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
