# Peer review of "Allied Health Professions Accreditation Standards for Work Integrated Learning: A Document Analysis"

_ijerph, 2023, doi:10.3390/ijerph20156478_

Round 1

Reviewer 1 Report

This study explored the similarities and differences in how Australian AHPs conceptualised WIL as evidenced through their accreditation requirements. To this aim, they screened curricula and standards, which were the object of thematic analysis. WIL is central to all the AHP, but the authors find that multiple definitions of WIL exist, which may make challenging for those who develop and/or review their profession’s accreditation documentation to develop valid standards. At the simplest level, WIL is described as being the bridge between theory and practice, but the authors argue that placements should be better conceptualised and assessment standards defined.

The study is original, and potentially useful for further regulating the AHP. However, there are a few details that should be revised.

In the introduction, it should be noted that positivist and constructivist stand for learning theories, not research methods. Moreover, there is no need to deepen on that.

As for the methods, the research design should not be a convenience technique that depends on the profile of the researchers.

Last, I have found interesting relationships with other fields, in which learning theory into practice has also a central role; for example, education. Although the profession is regulated, work placements are often ill-defined, just in terms of "competence", and they would benefit from a shift in perception, from a bridge between theory and practice to a place to go on learning. I suggest exploring these links, as it would make the article interesting for a wider readership.

Author Response

Reviewer 1 feedback

Our response

Adjustment to text

In the introduction, it should be noted that positivist and constructivist stand for learning theories, not research methods. Moreover, there is no need to deepen on that.

Thank you for your comment

To ensure clarity we have adjusted the introductory sentences and included additional references from to support points made.

Text has been inserted into Lines 137 to 141 which now reads:

The design of this study was underpinned by both positivist27 and constructivist1, 28 research paradigms. To answer the research questions, a non-experimental explanatory mixed methods research design29 consisting of two stages was selected to guide the collection and analysis of data obtained from publicly available AH professions documents.

the research design should not be a convenience technique that depends on the profile of the researchers.

Thank you for this comment. The reviewer may be referring to the methods section where we aim to communicate to the reader what may have influenced our approach to and implementation of the study. Additional statements have been included to outline our personal and collaborative interpersonal reflexivity as recommended by Olmos-Vega et al (2023)[1] for health profession education research using qualitative methods.

Should the editor determine that our positionality statement is not needed, we will remove the paragraph starting with “this duality of positioning…”

Text has been inserted in lines 150-153 and 163-166 which now reads:

This duality of positioning is also a reflection of the research team involved. As recommended by Olmos-Vega, et al30 for health profession education researchers, we next outline the professional backgrounds of each author which may have shaped decisions made throughout the project.

And:

In addition, the team members engaged in an ongoing process of collaborative interpersonal reflexivity to further explore the interplay of motivations, assumptions and expectations32 across the team and how these influenced decisions made throughout the research process.

Last, I have found interesting relationships with other fields, in which learning theory into practice has also a central role; for example, education. Although the profession is regulated, work placements are often ill-defined, just in terms of "competence", and they would benefit from a shift in perception, from a bridge between theory and practice to a place to go on learning. I suggest exploring these links, as it would make the article interesting for a wider readership

We agree with this reviewer’s comments and we present this in our discussion (final paragraph). We have added text to the discussion with reference to teacher education (Dean 2023) and other professions (Winchester- Seeto and Piggott, 2020).

However, if the reviewer has additional references that would add to this paragraph we are open to suggestions for further additions.

Text has been inserted in lines 586-595 (Paragraph 5 of the Discussion), which now reads:

Perspectives of WIL amongst other professions also demonstrate a range of learning that occurs through placement. In teacher education, WIL is observed to develop employability skills such as teamwork, communication and problem solving, and to pro-vide the foundation for lifelong learning.8 Similarly, Winchester-Seeto and Piggott92 propose a focus on ‘work-force’ learning, rather than ‘work-place’ learning, with ‘work-force learning including broader range of settings and learning activities such as virtual placements and simulation, to enhance students critical thinking and the ability to articulate professional identity. As the nature of work continues to change, these capabilities will more effectively prepare graduates for professional practice into the future.8, 92

[1] Olmos-Vega, F.M., Stalmeijer, R.E., Varpio, L., & Kahlke, R. (2023) A practical guide to reflexivity in qualitative research: AMEE Guide No. 149, Medical Teacher, 45(3), 241-251, https://doi.org/10.1080/0142159X.2022.2057287

Reviewer 2 Report

The presented work aimed at analyzing the education standards for WILL across AHPs in Australia is of great conceptual and methodological interest. In spite of this, some modifications are suggested below to improve quality in the opinion of this reviewer:

- It is recommended to include the authors' contributions at the end of the paper in a specific subsection, instead of including them at the beginning of the method section. 

- Clarify the use of abbreviations for the terms "allied health professions" and "allied health professional(s)" throughout the text. AHP, AHPs, AHPS are interchanged leading to reading difficulties (i.e. lines 521, 522, 561).

Author Response

Reviewer 2 comments

Our response

It is recommended to include the authors' contributions at the end of the paper in a specific subsection, instead of including them at the beginning of the method section. 

Thank you for your comment. Authors contributions are included at the end of the paper. However, with the growth of qualitative research in health professions education, authors are encouraged to communicate reflexivity to support the rigor of the research (Omos-Vega, Stalmeijer, Varpio & Kahlke, 2023)[1]. Additional statements have been included to outline our personal and collaborative interpersonal reflexivity

Should the editor determine that our positionality statement is not needed, we will remove the paragraph starting with “this duality of positioning…”

Text has been inserted in lines 150-153 and 163-166 which now reads:

This duality of positioning is also a reflection of the research team involved. As recommended by Olmos-Vega, et al30 for health profession education researchers, we next outline the professional backgrounds of each author which may have shaped decisions made throughout the project.

And:

In addition, the team members engaged in an ongoing process of collaborative interpersonal reflexivity to further explore the interplay of motivations, assumptions and expectations32 across the team and how these influenced decisions made throughout the research process.

Clarify the use of abbreviations for the terms "allied health professions" and "allied health professional(s)" throughout the text. AHP, AHPs, AHPS are interchanged leading to reading difficulties (i.e. lines 521, 522, 561).

Thank you for your feedback and on re-reading we agree that the use of AHP with/without the “s” has interchangable meanings.  We have adjusted the initialism to use only AH for Allied health and put the P in full to refer to profession/ professional/professionals

Numerous changes throughout the document.

[1] Olmos-Vega, F.M., Stalmeijer, R.E., Varpio, L., & Kahlke, R. (2023) A practical guide to reflexivity in qualitative research: AMEE Guide No. 149, Medical Teacher, 45(3), 241-251, https://doi.org/10.1080/0142159X.2022.2057287

Reviewer 3 Report

The article in general offers an interesting and practical theme for the solution of specific problems in areas of organizational learning and health. I would like to express some brief recommendations, which, in my opinion, would improve the document and which I submit to the authors' consideration:

a). The abbreviations mentioned in the abstract need to be defined again when they are used for the first time in the text (for example, WILL).

b). In the methodology, if they are not included, the following elements of the research design should be briefly mentioned: (1) paradigmatic approach (qualitative, quantitative or mixed); (2) nature (experimental or non-experimental): (3) purpose (exploratory, descriptive, correlational, deriving from structures or causal explanatory); and (4) temporality (longitudinal or transversal).

c). Try to mention in the text of the abstract the following elements: (1) problem studied; (2) study participants; (3) methodology used; (4) main findings; and (5) brief conclusions.

d). Is it possible to transfer Table 1 to an annex?

e). For me, the limitations of the study should be located after the conclusions. Also, briefly mention some prospects for future research on the subject based on this study.

Author Response

Reviewer 3 feedback

Our response

Changes made

 The abbreviations mentioned in the abstract need to be defined again when they are used for the first time in the text (for example, WILL).

Thank you for this feedback. We have reviewed the introduction to our article and note that the first time WIL is introduced, the definition of the term is introduced immediately in the following section 1.1. titled Definitions of WIL.

However, to assist the reader, we have moved the bracket from the end of the list of similar terms used and reordered the sentence so that WIL is introduced as the lead in sentence to definitions presented in Section 1.1.

The definitions have been expanded further in the first paragraph to aid the understanding of the reader.

Text has been inserted into Lines 53-58 which now reads:

On successful completion of their degrees, AHPS are deemed to be able to practice as semi-autonomous or autonomous practitioners. To support this capability, a core component of AHPs degrees is work integrated learning (WIL), also known as clinical education, fieldwork placement or practicum placements.

Text has been inserted into Lines 63-68, which now reads:

Further, the Tertiary Education Quality and Standards Agency states that WIL includes any “arrangement where students undertake learning in a work context as part of their course requirements”,7(p1) that is, “WIL describes strategies and activities that promote student learning through engaging with aspects of work”.8(p12) Thus, WIL is a pedagogical approach that promotes student learning and prepares graduates for their professions.8

In the methodology, if they are not included, the following elements of the research design should be briefly mentioned: (1) paradigmatic approach (qualitative, quantitative or mixed); (2) nature (experimental or non-experimental): (3) purpose (exploratory, descriptive, correlational, deriving from structures or causal explanatory); and (4) temporality (longitudinal or transversal).

Thank you for your feedback.

We have revised the first  paragraph of the methods chapter to include the reviewers comments.

We have outlined the

1) paradigmatic approach (qualitative, quantitative or mixed);

(2) nature (experimental or non-experimental): (3) purpose (exploratory, descriptive, correlational, deriving from structures or causal explanatory);

Text has been inserted into Lines 137 to 149 which now reads:

The design of this study was underpinned by both positivist27 and constructivist1, 28 research paradigms. To answer the research questions, a non-experimental explanatory mixed methods research design29 consisting of two stages was selected to guide the collection and analysis of data obtained from publicly available AH professions documents. Stage one proposes that the accreditation documents are the “source of truth” and therefore our answers to research question 1 could be extracted from these documents (positivist approach27) to collate and compare data. Stage 2 of the project uses a constructivist approach,1, 28 to analyse and interpret the conceptual understandings of WIL as articulated in the documents,29 This approach sought to understand and interpret the data to elicit the breadth of conceptual constructs across the professions, rather than compare findings (i.e. Research Question 2).

4. temporality (longitudinal or transversal).

We have indicated in the description of our approach to document analysis that the retrieved documents represented those that were current in each profession at a single point in time (February, 2023). We have also included to timeline for when data extraction was completed (March-April, 2023).

Text has been inserted into Lines 209-210 which now reads:

The retrieved documents represented those that were current as of February, 2023.

And Lines 215-217:

The data were then extracted from all documents and organised in a tabulated format in March-April, 2023.

Try to mention in the text of the abstract the following elements: (1) problem studied; (2) study participants; (3) methodology used; (4) main findings; and (5) brief conclusions.

Thank you for this feedback. We have reviewed the abstract and believe that we have addressed the recommended elements.

The abstract now reads:

A key role of allied health (AH) professional regulatory and professional bodies is to ensure that AH education programs provide work integrated learning (WIL) opportunities for students. The requirements are outlined via the respective profession's educational accreditation standards. Although a significant component of the AH professional degrees, researchers have not explored how standards specific to WIL are developed, nor how WIL might be conceptualised through the standards.  This study explored how WIL is conceptualised through comparing the WIL education standards across Australian AH professions. Using a non-experimental explanatory mixed methods research design, a document analysis of Australian education program accreditation standards (and associated documents) for 15 AH professions was undertaken. Data analysis included inductive textual and thematic analyses to compare AH professional conceptualization of WIL. This study found a high degree of variation in how AH professions describe WIL. While there was a common requirement for students to demonstrate competency in WIL, requirements for WIL quantity, assessment and supervision varied. Four key themes were identified regarding the contribution of WIL to curriculum and student learning: 1) the relationship of WIL to the program curriculum; 2) WIL as a learning process; 3) learning from diverse WIL contexts; and 4) developing competence through WIL. (4) Conclusion: Overall, the diversity in the standards reflected differing understandings of what is WIL. Thus in the absence of frameworks for designing accreditation standards, the risk is that some AH professions will continue to perpetuate the myth that WILs primary purpose is to be the bridge between theory and practice.

Is it possible to transfer Table 1 to an annex?

Thank you for this suggestion which we have considered. We believe that to remove the Table to a supplementary file may influence the credibility of our work. We acknowledge the table is lengthy, but it provides a summary of professions included for the data collection and the documents sourced.

However, we will take the editor’s advice about the location of the table related to page or table limits.

No change made to location of Table.

For me, the limitations of the study should be located after the conclusions.

Thank you for this comment. We have reviewed other articles in the same special issue and all place Limitations before the conclusions.

No change made to the order of the section.

Also, briefly mention some prospects for future research on the subject based on this study.

Thank you for this comment. We have included indicators for future research in the limitation and conclusion sections

Text has been inserted into Lines 626-633 which now reads:

Finally, as this study specifically focused on the conceptualisation of WIL within the accreditation standards for each profession, the findings do not represent the various conceptualizations of WIL that may have been drawn from the written curriculums of professional education programmes or more directly from curriculum developers, practicing professionals, or the students undertaking WIL as part of their education. Further research into the perspectives of other stakeholders might illuminate very different conceptualisations of the purpose, value and learning outcomes of WIL.

And in the conclusion line 656 - 659:

However, to fully understand what these are, each AH profession needs to consider how they conceptualise placements and how this is then articulated through their accreditation standards. Further research is needed.

Reviewer 4 Report

Dear authors, the work presented still presented an important bias from the beginning in the selection of the material. A review of the literature should be carried out according to the prism statement. I have not observed that a clear answer to the research question is given after reading the manuscript several times. The analysis is superficial and does not follow a research model that allows me to reproduce it to verify the veracity and validity of the results.

It needs a thorough review and focus by the authors.

Author Response

Reviewer 4 feedback

Our response

Dear authors, the work presented still presented an important bias from the beginning in the selection of the material.

Thank you for this comment.  We are unsure as to how to interpret the comment that important bias was evident in the selection of the material. Our study purposely focused on how Australian accrediting or regulatory authorities express their WIL requirements in their accreditation documentation. We outline the process by which the professions to be included was determined by citing the references used to inform the initial list and our exclusion/inclusion criteria.

However, we have added a paragraph to the limitations with reference to further research from the perspective of other stakeholders.

Text has been inserted into Lines 629-636 which now reads:

Finally, as this study specifically focused on the conceptualisation of WIL within the accreditation standards for each profession, the findings do not represent the various conceptualizations of WIL that may have been drawn from the written curriculums of professional education programmes or more directly from curriculum developers, practicing professionals, or the students undertaking WIL as part of their education. Further research into the perspectives of other stakeholders might illuminate very different conceptualisations of the purpose, value and learning outcomes of WIL.

A review of the literature should be carried out according to the prism statement.

It was not the intent of this paper to undertake a systematic or similar review of the published literature. The recommendation to review the literature according the PRISM[A} statement does not appear to be relevant to the focus of our paper.

No changes made

“not observed that a clear answer to the research question is given

Thank you for your feedback. On review we have more clearly linked our conclusion to the study aims and findings.

Text has been inserted into Lines 640-662 which now reads:

Whilst the whole degree contributes to the readiness of the AH professional to practice semi-autonomously or autonomously, the findings of this study demonstrate that the WIL components of the degree contribute significantly to the student’s progress towards being that work-ready graduate. The importance of this is highlighted in the education accreditation standards, with sections often “set-aside” with specific standards related to WIL. This study aimed to understand the similarities and differences between Australian AH professions in their WIL requirements, and found multiple terms and definitions of WIL exist, with inconsistent requirements for the amount and type of settings. This finding suggests that it may be challenging for those who develop and/or review their profession’s accreditation documentation to develop valid standards. Whilst frameworks for the development of competencies exist, the same does not appear to be available for accreditation standards and the process by which accreditation standards are developed may only be included in the accreditation documentation and thus not subject to critical peer review.

The study also aimed to discover how Australian AH professions conceptualise WIL in their accreditation requirements. Across the 15 AH professions, we found diverse understandings of WIL leading us to question how WIL is being conceptualised. At the simplest level, WIL is described as being the bridge between theory and practice. However, we argue that to only consider WIL as this, is to do a disservice to what WIL actually offers. As a crucible for student learning, WIL provides rich learning experiences. However, to fully understand what these are, each AH profession needs to consider how they conceptualise placements and how this is then articulated through their accreditation standards. Further research is needed.

analysis is superficial

Thank you for the feedback. The focus of this mixed methods study based on the written text within the accreditation standards documents did not permit in-depth analysis. Document analysis processes were used as described. The sparsity of textual information related to WIL in the documents limited the qualitative analysis of the data, however the process was consistent with Braun and Clarke’s (2022) methods, and the quantity and quality of the data provided in accreditation standards.

No change

does not follow a research model that allows me to reproduce it to verify the veracity and validity of the results.

We believe that we have provided a model that allows reproduction or replication of the research.

We have outlined:

·      Our research aims

·      Our inclusion/exclusion criteria

·      Demonstrated the source of the data

·      Cited the document analysis process which is well-recognised and used in the literature

·      Outlined our protocol for data extraction and data analysis

·      Used tables to document the data extracted from the tables.  If a reader chose to replicate our method, they would be able to compare the data they extracted to ours to confirm the veracity of the results. 

We do note that the tables in the appendices appear to have been omitted from the end of the article and this may have led to the reviewer’s comment. These will be uploaded as supplementary files for the review.

No changes made

Round 2

Reviewer 4 Report

The changes, although related, do not conform to a minimum, the selection of the material based on this bias is still not clear. I do not consider myself in a position to approve this scientific work.

Author Response

Thank you to Reviewer 4 for their second review of our manuscript. Our responses to the two points raised are as below:

Point 1:  The changes, although related, do not conform to a minimum, the selection of the material based on this bias is still not clear.

Response 1: We have considered your assessment/comments and are unsure of how to proceed with revisions.  For example, it would be helpful for us to know what is meant by "conforming to a minimum and the selection of material based on this bias is still not clear". For example, 
• is there additional background and relevant references you had expected to be in the Introduction? If so, could you please indicate what this might be?  
• is there sections in the methods that you would like to see more detail for? If so, please indicate what this might be?

Point 2:
The Academic editor noted that Reviewer 4 raises an important issue that there still may be about 34% redundancy with other published work

Response 2: 

We believe this may have occurred related to our use of 15 allied health professional regulatory authority documents that were reviewed for our study. These documents are all listed in Table 1 and included in the references for the manuscript. In addition, to support our interpretation, we have included a number of quotes from these documents throughout the results section of our manuscript. All citations have been included in the reference list and direct quotes are cited with page numbers. We believe this should explain the higher-than-expected redundancy percentage.

No changes have been made to the manuscript in relation to Reviewer 4's report. Minor editorial changes have been made to the manuscript (i.e. adjusting referencing, minor grammatical errors).